# Enhancing the Contact Area of Ti Wire as Photoanode Substrate of Flexible Fiber-Type Dye-Sensitized Solar Cells Using the TiO_2_ Nanotube Growth and Removal Technique

**DOI:** 10.3390/nano9111521

**Published:** 2019-10-25

**Authors:** Mao-Shen Tien, Lu-Yin Lin, Bing-Chang Xiao, Siao-Ting Hong

**Affiliations:** 1Department of Chemical Engineering and Biotechnology, National Taipei University of Technology, Taipei 10608, Taiwan; okl445m9@gmail.com (M.-S.T.); xerioe86507@gmail.com (B.-C.X.); luyo0815@gmail.com (S.-T.H.); 2Research Center of Energy Conservation for New Generation of Residential, Commercial, and Industrial Sectors, Taipei 10608, Taiwan

**Keywords:** anodization, fiber-type dye-sensitized solar cell, intensity modulated photocurrent spectroscopy/intensity modulated photovoltage spectroscopy, TiO_2_ nanotube, Ti wire

## Abstract

The fiber-type dye-sensitized solar cell (FDSSC) with flexible and dim-light workable features is one of the promising energy generation devices for soft electronics. A novel TiO_2_ nanotube (TNT) growth and removal technique is proposed in this study to enhance the contact area of the Ti wire substrate using anodization and ultrasonication processes. Smaller and denser imprints of TNT on the surface of Ti wire are obtained when a smaller voltage was applied for anodization. The thickness of the TiO_2_ nanoparticle layer coated on the Ti wire is also optimized by varying the dip-coating layers. With the smallest diameter and densest distribution of TNT imprints on the Ti wire, the FDSSC with the TiO_2_/TNT-printed Ti wire photoanode, prepared using 30 V as the anodization voltage, shows the highest photon-to-electricity efficiency of 2.37% as a result of the rough surface of Ti wire substrate, which provides more contact, as well as the suitable thickness of the TiO_2_ nanoparticle layer, which promotes charge generation and transportation. The smallest charge-transfer resistance and the highest electron collection efficiency are also obtained in this case, as examined using the electrochemical impedance spectroscopy and intensity modulated photocurrent spectroscopy/intensity modulated photovoltage spectroscopy. This facile TNT growth and removal technique is expected to be able to be applied to other fields for enhancing the contact area of the titanium substrate and promoting the generation of electrochemical reactions.

## 1. Introduction

Soft electronics are developed intensively nowadays to facilitate safer and more comfortable lives for modern people [1,2,3]. Hence, the energy supply of soft electronics of great significance for determining the sustainability and feasibility of devices in real applications. The dye-sensitized solar cell (DSSC), with its low cost and non-toxic properties, has been considered as one of the promising energy generation devices for soft electronics [4,5,6]. Good operational ability under dim-light and indoor conditions promotes the application of DSSC for soft electronics which are required to be used in indoor conditions [7,8,9]. 

The photoanode of the DSSC plays the most important role in light absorption, charge generation, and transportation [10,11,12,13]. The surface area of the current collector is highly important for determining the contact between the substrate and semiconductor as well as the transfer of charge [14,15,16]. Hydrogen peroxide has been widely used to increase the surface area of metal substrates. Tsai et al. applied H_2_O_2_ pretreatment on Ti foils to form networked TiO_2_ nanosheets to enhance the surface area of the Ti foil [16]. Lee et al. prepared a sponge-like and conformal TiO_2_ underlayer by using hydrogen peroxide oxidation [17]. An et al. prepared a TiO_2_ nanoforest underlayer on a Ti substrate using acid and H_2_O_2_ treatments for improving the photovoltaic performance of a Ti substrate-based DSSC with higher surface roughness of substrate and better electrical contact between Ti substrate and TiO_2_ nanoparticles (TNP) [18]. Linnemann et al. used a HNO_3_ treatment to increase the electrochemically active surface area of the TiO_2_ layer on Ti foil. The photovoltaic performance of the pertinent DSSC was hence improved [19]. Bialuschewski et al. applied a femtosecond laser structuring technique on titanium substrate to enhance surface area, and overall photoelectrochemical water splitting performance was improved with the treated titanium substrate [20]. Zahran et al. used different amounts of hydrofluoric acid to etch titanium substrates and studied surface area enhancements [21]. However, extra TiO_2_ nanomaterials should be coated on the current collector and, hence, more interfaces could be developed to increase charge-transfer resistance. In our previous work, the current collector was modified using the TiO_2_ nanotube (TNT) growth-removal process [22,23]. The TNT can also be used as the dye-adsorbing layer in other works [24,25]. The modified Ti foil was applied for sputtering Pt as the counter electrode for the normal-type DSSC. Due to the larger surface area of substrate for the counter electrode, a better catalytic ability for redox reactions in electrolyte was achieved and, hence, higher power conversion efficiency was obtained for the resulting DSSC. Based on this concept, the surface area of Ti wire and hence contact between the TNP layer and the Ti wire substrate are expected to be enhanced by using the unique TNT growth-removal technique for facilitating charge transfer. 

In this work, the Ti wire was treated using the TNT growth-removal process and then used as a substrate for depositing TNP via the facile dip-coating process. Different anodization voltages were applied to fabricate different sizes of TNT imprints on the Ti wire surface. Electrochemical impedance spectroscopy and intensity modulated photocurrent spectroscopy/intensity modulated photovoltage spectroscopy (IMPS/IMVS) were also applied to analyze charge-transfer resistance and collection efficiency of fiber-type dye-sensitized solar cell (FDSSC) to further understand effects of the TNT-growth-removal treatment on the photovoltaic performance of FDSSCs.

## 2. Experimental

### 2.1. Pretreatment of Ti Wires Using the TiO_2_ Nanotube Growth and Removal Process and Coating of TiO_2_ Nanoparticles on Ti Wires

Ti wires (diameter = 0.5 mm, 99.5%, Taiwan) were washed using neutral cleaner, deionized water (DIW), and acetone under sonication in sequence. The neutral cleaner was used to wash the substrate via removing the organic materials. The pretreatment of the growth and removal process was carried out by firstly anodizing Ti wire using a power supply with a Pt wire as the counter electrode, and then removing TNT from Ti wire surface by ultrasonicating the anodized Ti wires in DIW. The TNT-printed Ti wires were thereby obtained.

The TNP layer was then coated on the as-obtained Ti wire and the TNT-printed Ti wire by using a dip-coating technique with a withdrawal rate of 100 mm/min. The thickness of the TNP layer plays an important role in the dye adsorption and charge recombination. Hence, prior to applying the Ti wires treated with the TNT growth and removal process as the substrate for the photoanode of the FDSSC, the thickness of the TNP layer on the as-obtained Ti wire was first optimized by using different dip-coating times. The TNP paste contains 6 g commercial TiO_2_ (P25, Taiwan), 2 mL acetyl acetone, and 0.1 mL Triton X-100 (VETEO, Taiwan) in 20 mL ethanol (99.5%, ECEO, Taiwan). The TNP-coated Ti wire photoanode was then annealed at 450 °C for 30 min to improve crystallinity of TiO_2_ and enhance contact between the TNP and Ti wire.

### 2.2. Assembly of Fiber-Type Dye-Sensitized Solar Cells

The TNP/Ti wire and TNT-printed Ti wire were immersed in 0.3 mM N719 (99%, UniRegion Bio-Tech, Taiwan) solution with a mixed solvent of dehydrated acetonitrile/tert-butanol mixture solvent (vol. ratio 1:1) overnight to fabricate dye-coated TNP/Ti wire photoanodes. The FDSSC was assembled using a dye-coated flexible TNP/Ti wire photoanode and a Pt wire counter electrode which were put into a glass capillary with electrolyte fully injected. The glass capillary was 1.5 mm in diameter and of 3 cm long. Both electrodes were flexible but the glass capillary was not. By replacing the liquid electrode with the gel electrolyte and removing the glass capillary from the device, the whole device was thus flexible. The electrolyte contains 0.5 M lithium iodide (LiI, 99%, ACROS, Taiwan), 0.05 M iodine (I_2_, 98%, TCI, Taiwan), 0.5 M 1,2-dimethyl-3-propylimidazolium iodide (DMPII, 98%, TCI, Taiwan) and 0.5 M 4-tertbutyl-pyridine in dehydrated acetonitrile (TBP, 99%, J.T. Baker, Taiwan). Finally, hot melt adhesive was used to seal the tubular container at both ends to complete the assembly of the FDSSC. 

### 2.3. Material Characterizations and Electrochemical Measurements

The photovoltaic performance of FDSSCs was measured using the potentiostat/galvanostat instrument with an FRA2 module (PGSTAT 204, Autolab, Eco Chemie, the Netherlands) under illumination of solar simulator (X500, BLUE SKY, Taiwan) with irradiance of 100 mW cm^−2^ at an equivalent air mass (AM) of 1.5. All the data were collected using three devices for each anodized sample. This technique is repeatable and reliable. Electrochemical impedance spectroscopy (EIS) measurement was conducted at open-circuit potential and with frequencies between 0.01 Hz to 100 kHz. Intensity modulated photocurrent/photovoltage spectroscopy (IMPS/IMVS) were conducted at an electrochemical workstation with light emitting diodes (LED) driven by a source supply. Field emission scanning electron microscopy (FE-SEM, FEI Nova230, Taiwan) was applied to observe morphology of TiO_2_/Ti wires, and X-ray diffraction patterns (XRD, X’Pert^3^ Powder, PANalytical, Taiwan) were used to analyze the composition of TiO_2_/Ti wires.

## 3. Results and Discussion

The thickness of the TNP layer was firstly examined from the SEM images. The side-view SEM images for the TNP/Ti photoanodes prepared using one, two, three, four and five dip-coating layers are shown in Figure 1a–e. It was found that a thicker TNP layer could be obtained by using more dip-coating layers to fabricate the photoanodes. The photovoltaic performance of the FDSSC with TNP/Ti photoanodes prepared using different dip-coating layers was further evaluated using linear sweep voltammetry (LSV) curves, as shown in Figure 1f. To compare the photovoltaic parameters relating to the thickness of the TNP layer more clearly, trends of open-circuit voltage (*V*_OC_), photocurrent density (*J*_SC_), and power conversion efficiency (*η*) of the FDSSC to TNP dip-coating layers on photoanode was presented in Figure 1g. The *V*_OC_ and *J*_SC_ values of the FDSSC increased with thicker TNP layers and achieved the largest values with three TNP layers. Thicker TNP layer could be obtained when more TNP layers were coated on the Ti wire using a dip-coating process. The increases in the *J*_SC_ value for the FDSSC with thicker TNP layer is attributed to more electron excitation with more dye molecules adsorbed on thicker TNP layers. More charge could be accumulated in the conduction band of TiO_2_ with thicker layers, so the difference between the charge-accumulated conduction band edge of TiO_2_ and the redox potential of the electrolyte was larger, leading to increased *V*_OC_ values for the FDSSC with the photoanode containing three TNP layers compared to the FDSSC with the photoanode containing fewer TNP layers. However, further increasing TNP layers on the photoanodes lead to reductions on both *V*_OC_ and *J*_SC_ values. This is due to the TNP layer being thicker than the charge diffusion length in the photoanode of FDSSC. When the thickness of the TiO_2_ layer is greater than the charge diffusion length, serious recombination may occur since photon-generated charges may be unable to reach such a thick current collector. Based on this optimized thickness of TNP layer, modified Ti wires were further applied as the substrate of photoanode for FDSSC. The physical properties of photoanodes and electrochemical performance of FDCCSs were further analyzed. 

SEM images for Ti wires and TNP-coated Ti wires were examined, as shown in Figure 2. Figure 2a–d respectively presents the as-obtained Ti wire and the pretreated Ti wires using anodization voltages of 30, 40, and 50 V. The SEM images for the anodized Ti wire without the TNT removed were inserted in the corresponding figure in Figure 2b–d. The as-obtained Ti wire shows a smooth surface with some randomly distributed scratches on surface. The Ti wires treated using the TNT growth and removal process with different anodization voltages show uniform circular inks on the surface. Since the TNT has circular open ends, the removal of TNT would leave circular shapes on the Ti wire. Hence, the term “circular inks” was used to describe the remaining shapes on the Ti wire. The diameter of the circular inks on the Ti wire surface increased when higher anodization voltages were applied in the TNT growing process. The average diameter of 91.0, 136.5, and 182.3 nm were respectively obtained for the circular inks on Ti wire surface prepared using 30, 40, and 50 V for anodization. The size of circular inks is inferred to be similar to that of the TNT grown on Ti wire. Furthermore, SEM images of TNP/Ti wires with substrate of the as-obtained Ti wire and the pretreated Ti wires using anodization voltages of 30, 40, and 50 V are shown in Figure 2e–h. The high magnifications for the SEM images are shown in the corresponding figures. Due to the TNP coating, all electrodes show similar images with several nanoparticles distributed. Based on the similar thickness of TNP layers prepared using the same dip-coating process, different Ti wire substrates are inferred to mainly influence the contact between substrate and TNP layer coated on surface of Ti wire. The influence of the Ti wire surface configuration on charge transfer in the TNP layer far from substrate is considered to be limited.

The preparation of Ti wires using the TNT growth and removal technique was further examined from the XRD pattern (Figure 3a). Pure titanium metal signal was observed in the pattern of as-obtained Ti wire. After carrying out the anodization process for Ti wire, the anodized Ti wire shows anatase TiO_2_ peaks in the pattern, indicating formation of TNT on the surface of the Ti wire. Removal of TNT was conducted by sonicating anodized Ti wires in the DIW and the peaks of anatase TiO_2_ totally disappeared in the XRD pattern. This phenomenon suggests the successful removal of TNT from the surface. Further coating the TNP layer on the sonicated anodized Ti wire leads to the appearance of anatase TiO_2_ peaks, which is attributed to the TNP layer. The result of the XRD pattern implies the successful fabrication of pure Ti wire substrate using the TNT growth and removal technique, since there are no extra peaks of titanium metal for the sonicated anodized Ti wire. Furthermore, the composition of TNP/Ti electrodes was examined from the XRD pattern in Figure 3b, where the standard pattern of anatase TiO_2_ was also shown for comparison. The as-obtained Ti wire is indicated as Ti (0 V) in this figure. Again, the pure titanium peaks were observed in this pattern. With the coating of the TNP layer, the TNP/Ti (0 V) shows consistent peaks of anatase TiO_2_ owing to the anatase phase nature of the TNP layer. 

The photovoltaic performance of FDSSC with TNP/Ti wire and TNP/TNT-printed Ti wire photoanodes was further analyzed from LSV curves, as shown in Figure 4. Photovoltaic parameters were listed in Table 1 for clearer comparison. The as-obtained Ti wire without anodization is indicated as Ti (0 V) in this figure. *V*_OC_ values are similar for all FDSSCs, suggesting the recombination rate is similar for the FDSSC with different substrates for photoanodes. The higher *J*_SC_ values were obtained for the FDSSC with the TNP/TNT-printed Ti wire photoanodes compared to that for the cell with the TNP/Ti wire photoanode, owing to the TNT imprints on the substrate of the photoanode providing better charge transfer and larger surface area for dye adsorption [22]. The *FF* values are also higher for the FDSSC with TNP/TNT-printed Ti wire photoanodes, indicating the reduced charge-transfer resistance for these cases with modified substrate for photoanodes. On the other hand, among the FDSSC with TNP/TNT-printed Ti wire photoanodes, the highest *J*_SC_ of 4.66 mA/cm^2^ and *η* of 2.37% were achieved for the cell with photoanode substrate anodized with 30 V. It is inferred that the smallest diameter for the TNT imprints prepared using 30 V could provide the largest surface area for TNP deposition and more routes for transferring charges [22]. Therefore, the best photovoltaic performance was obtained for the FDSSC with the photoanode prepared using substrate pretreated with the smallest anodization voltage. The variation of the anodization voltage cannot only influence the diameter but also the depth of TNT inks. When using an anodization voltage smaller than 30 V, almost no inks can be observed in the SEM images. Therefore, 30 V is the smallest acceptable anodization voltage to create TNT inks on the Ti wire in this study. 

Furthermore, the charge-transfer resistance in the photoanode of the FDSSC with the TNP/Ti wire and TNP/TNT-printed Ti wire photoanodes was examined using a Nyquist plot, as shown in Figure 5a. The equivalent circuit for fitting charge-transfer resistances of the FDSSC was also included in this figure. The charge-transfer resistance at the interface between electrolyte and dye/TiO_2_/Ti wire (*R*_ct2_) which was evaluated by fitting a semicircle in the middle frequency region was also listed in Table 1. The largest *R*_ct2_ value of 155 Ω was obtained for the FDSSC with TNP/Ti wire photoanode, probably due to the lack of TNT imprints on the current collector reducing contact between the substrate and TNP layers. On the other hand, the *R*_ct2_ value decreases for the FDSSC with the TNP/TNT-printed Ti wire photoanode prepared using smaller voltages for anodization. The smallest *R*_ct2_ value of 97 Ω was obtained for the FDSSC with the TNP/TNT-printed Ti wire photoanode prepared using 30 V for anodization. Since the smaller diameter of the TNT imprints can be obtained on the Ti wire anodized using smaller voltage, the smallest diameter of the TNT imprints for the Ti wire anodized using 30 V is favorable for providing the most contacts between substrate and TNP. In addition, the electron lifetime in the FDSSC with TNP/Ti wire and TNP/TNT-printed Ti wire photoanodes was examined by from Bode plot, as shown in Figure 5b. The electron lifetime is inversely proportional to the frequency for the peak at low frequency regions. The FDSSC with the TNP/TNT-printed Ti wire photoanode prepared using 30 V for anodization shows the smallest frequency, indicating the longest electron lifetime for this case. Similarly, the most contacts between current collector and TNP can provide the most sites for charge transfer from the electron generating points to the outer circuit. The longest electron lifetime for the FDSSC with the TNP/TNT-printed Ti wire photoanode prepared using 30 V for anodization is due to the smallest TNT imprints on Ti wire with the most contacts with TNP for providing the most chances for transporting charges. The smallest charge-transfer resistance and the longest electron lifetime for the FDSSC with the TNP/TNT-printed Ti wire photoanode prepared using 30 V are consistent with its highest *η* value as analyzed from LSV curves in Figure 4. This result suggests that treatments of anodization and ultrasonication for fabricating TNT imprints are beneficial for enhancing the transportation of charges and prolonging the survival of charges in the FDSSC during the illuminating process. Last, electron transfer behavior of the FDSSC with the TNP/Ti wire photoanode and the TNP/TNT-printed Ti wire prepared using 30 V was further analyzed from IMPS and IMVS spectra, as respectively shown in Figure 6a,b. By evaluating electron transport time and electron lifetime, the electron collection efficiency for the FDSSC with the TNP/Ti wire and TNP/TNT-printed Ti wire photoanodes was calculated [26], as shown in Figure 6c. A higher electron collection efficiency was obtained for the FDSSC with TNP/TNT-printed Ti wire photoanode, as compared with that for the FDSSC with the TNP/Ti wire photoanode, due to more contact between the substrate and TNP layer to promote electron collection in the former cases. The higher electron collection efficiency of 84% was obtained for the TNP/TNT-printed Ti wire photoanode prepared using 30 V for anodization, while the cell with TNP/Ti wire photoanode only presented an electron collection efficiency of 60%. This result is again due to the smallest diameter of TNT imprints on Ti wire anodized using the smallest voltage of 30 V for providing more contact points for collecting charges. To more clearly explain the enhanced contact points for charge transfer for the Ti wire treated with anodization and ultrasonication, Scheme 1a,b respectively presents illustration of the charge transfer in FDSSC with the photoanode based on the as-obtained Ti wire and the Ti wire treated with the TNT growth and removal process. With TNT imprints on Ti wire, more contacts between current collector and TNP layer could be obtained and hence more efficient charge transfer could also be achieved for this case. TNT imprints are verified to play important roles on charge transfer in photoanode, and the smaller diameter of TNT imprints is inferred to be more preferable for enhancing contact points between Ti wire and TNP. The study on the design of TNT imprints is expected to further develop to achieve better photovoltaic performance of the FDSSC with flexible Ti wires as the photoanode substrate.

## 4. Conclusions 

The TNT growth and removal process was firstly proposed to enhance surface area of Ti wire as a substrate for the photoanode of FDSSC. Different anodization voltages were applied for developing TNT imprints on Ti wire with different diameters. The FDSSCs with the photoanode composed of TNT-printed Ti wire substrate show better photovoltaic performances compared to the cell with untreated Ti wire as its photoanode substrate. The highest *η* value of 2.37% along with the *V*_OC_ value of 0.70 V, *J*_SC_ value of 4.66 mA/cm^2^, and *FF* of 0.76 were obtained for the FDSSC with the optimized TNT-printed Ti wire as photoanode substrate prepared using 30 V as anodization voltage, owing to the smallest diameter of TNT imprints, in this case providing the most contact points between the substrate and TNP layer and to promote charge transfer. The smallest charge-transfer resistance, longest electron lifetime, and highest electron collection efficiency of 84% were also obtained for the FDSSC with the photoanode substrate prepared using 30 V as the anodization voltage. This work proposed a simple method to improve contact between the current collector and semiconductor layer of the photoanode of FDSSC. More effective substrates are expected to be developed by carefully designing an anodization program for creating TNT imprints on Ti wire with more efficient contact between subtracts and dye-adsorption layers.

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
