# Peer review of "Enhancing the Contact Area of Ti Wire as Photoanode Substrate of Flexible Fiber-Type Dye-Sensitized Solar Cells Using the TiO2 Nanotube Growth and Removal Technique"

_nanomaterials, 2019, doi:10.3390/nano9111521_

Round 1

Reviewer 1 Report

The article presents the TNT growth and removal process to enhance the surface area of Ti wire as the substrate for the photoanode of FDSSC.  Different anodization voltages were applied to fabricate different sizes of the TNT imprints on the Ti wire surface.

The manuscript is well written, there are only several minor issues to be addressed:

-In the Introduction part, the final paragraph must to be rewritten in a such way to describe just the purpose of this work. The best results obtained must to be introduced just in abstract or in conclusions.

-Also, the first paragraph from the Results and discussion part must to be introduced in Experimental part.

-In my opinion, a table with the all prepared samples, from the beginning at the end can be useful.

-Also, the SEM images of the TNP/Ti wire and the TNP/TNT-printed Ti wire prepared using 30, 40 and 50 V but at a higher magnification can be relevant.

Author Response

Reviewer #1

Specific comments

Comment 1: The article presents the TNT growth and removal process to enhance the surface area of Ti wire as the substrate for the photoanode of FDSSC. Different anodization voltages were applied to fabricate different sizes of the TNT imprints on the Ti wire surface. In the Introduction part, the final paragraph must to be rewritten in a such way to describe just the purpose of this work. The best results obtained must to be introduced just in abstract or in conclusions.

Response to specific comment 1: Thanks the reviewer for the suggestion. The best results obtained in this work were deleted in the last paragraph of the introduction part.

Comment 2: Also, the first paragraph from the Results and discussion part must to be introduced in Experimental part.

Response to specific comment 12: Thanks the reviewer for the suggestion. The first paragraph of the Results and discussion part was moved to the Experimental part colored in the yellow background in lines 12-16 at page 5 in the revised manuscript as follows.

“The thickness of the TNP layer plays important roles on the dye adsorption and charge recombination. Hence, prior to apply the Ti wires treated with the TNT growth and removal process as the substrate for the photoanode of the FDSSC, the thickness of the TNP layer on the as-obtained Ti wire was firstly optimized by using different dip-coating times.”

Comment 3: In my opinion, a table with the all prepared samples, from the beginning at the end can be useful.

Response to specific comment 3: Thanks the reviewer for the suggestion. Table 1 in the original manuscript already lists the photovoltaic parameters for all the FDSSC with the TNP/Ti wire and TNP/TNT-printed Ti wire.

Comment 4: Also, the SEM images of the TNP/Ti wire and the TNP/TNT-printed Ti wire prepared using 30, 40 and 50 V but at a higher magnification can be relevant.

Response to specific comment 4: Thanks the reviewer for the suggestion. The higher magnified SEM images for the TNP/Ti wire and the TNP/TNT-printed Ti wire prepared using 30, 40 and 50 V were provided in the inserted figures in Figure 2(e)-(h), respectively. The following sentences were also added in the line 4 at the bottom at page 10 in the revised manuscript colored in the yellow background to explain the inserted figures.

“The high magnifications for the SEM images were inserted in the corresponding figures.”

Reviewer 2 Report

The authors present their findings on their modified treatment on TiO2 for DSSCs. While the results are encouraging I believe the manuscript requires some clarifications on the scientific aspect. In addition, the manuscript needs proof-reading concerning the phrasing and writing style.

Concerning the scientific part. While this process is introduced by the authors (ref 24 and 24 for counter electrodes) this manuscript transfers this knowledge to the active layer of the photoanode.

It would also be good to mention other researchers working on DSSCs with TiO2 wires (e.g. ACS Nano, 2010, 44, 2196-2200; J. Mater. Chem., 2007,17, 1451-1457)

While in principle this technique is interesting and potentially useful for FDSSCs a few issues need to be addressed before publication.

a) Does this technique offer repeatability and reliability? How many devices were fabricated from each anodized sample?

I apologize if I have missed this but are the devices flexible? There is only one mention of this on line 104 "which were put into a glass capillary with electrolyte fully injected." What are the typical sizes of the glass capillary to accommodate the working and counter electrode?

b) line 89: What is a "neutral cleaner"?

c) line 107: Is the hot adhesive "Surlyn" or have the authors used another sealant?

d) lines 133-135: "The more charges generation may provide larger driving force for charge transfer and the recombination possibility may be reduced at the same time"  Why would the recombination be reduced? Such specific reasoning as reduced recombination requires additional experiments as this is a huge issue with solar cells and simple "more charge generation" cannot be a viable explanation. Moreover, on the solar cell with 4 layers (Figure 1a, green line) the Jsc is significantly lower but the Voc remains the same. Therefore, this does not hold true in that case. I understand that the authors are speculating on the issue of increased Voc but they should be more careful with assumptions.

e) line 154: What are "circular inks" Are they simply describing the morphology of the electrode after treatment?

f) line 200-202: "higher JSC values were obtained for the F DSSC with the TNP/TNT TNT-printed Ti wire photoanodes than that for the cell with the TNP/Ti wire photoanode, owing to the TNT imprints on the substrate of the photoanode to provide better charge transfer."  Why is that the explanation specifically?
What proves the improved charge transfer? Is this some results from earlier work from the authors? If so a reference is needed. Higher Jsc could simply be from improved dye absorption (higher quantity of dye due to more surface area).  

g) line 208-210: "Therefore, the best photovoltaic performance was obtained for the FDSSC with the photoanode prepared using the substrate pre-treated with the smallest anodization voltage."    The smallest anodization voltage measured here is zero (Fig 4) which does not show this behavior. Also since the authors see that the anodization voltage is inversely proportional to the efficiency, have they tried 20V, 10V? etc

Concerning the english language there are two main issues.

1) there are too many instances of the overusing the article "the" by the authors throughout the manuscript.

2) The authors tend to unnecessarily write extremely long sentences. There are several instances where the sentence is 6 or more lines long. This forces the interested reader to go back several times just to understand a simple point that the authors are trying to make. examples of unnecessarily long or confusing sentences include:

lines 27-33

lines 76-81

There are also bad sentences that definitely need rephrasing due to bad English or simplification such as

line 150-152

lines 249-252

Several simple corrections like this can greatly improve the interested reader's understanding of the authors' research.

Other than those two main issues the manuscript is acceptable in english with minor spelling mistakes. (e.g. 1st line of the abstract : "deem-light". I am assuming that the authors mean dim-light? It would be easier for the scientific crowd to write "excellent low-light performance" which is something often used to characterize DSSCs )

Unfortunately there are several minor phrasing errors which can be easily avoided by a thorough proof reading by an native english speaker.

Author Response

Reviewer #2

General comment: The authors present their findings on their modified treatment on TiO2 for DSSCs. While the results are encouraging I believe the manuscript requires some clarifications on the scientific aspect. In addition, the manuscript needs proof-reading concerning the phrasing and writing style. Concerning the scientific part. While this process is introduced by the authors (ref 24 and 24 for counter electrodes) this manuscript transfers this knowledge to the active layer of the photoanode. It would also be good to mention other researchers working on DSSCs with TiO2 wires (e.g. ACS Nano, 2010, 44, 2196-2200; J. Mater. Chem., 2007,17, 1451-1457) While in principle this technique is interesting and potentially useful for FDSSCs a few issues need to be addressed before publication.

Response to the general comment: Thanks the reviewer for the positive comment. The papers mentioned by the reviewer were cited in ref. 26 and 27 in the introduction part in lines 7-8 at page 4 in the revised manuscript colored in the yellow background as follows.

“The TNT can also be used as the dye-adsorbing layer in other works [26, 27].”

Comment 1: Does this technique offer repeatability and reliability? How many devices were fabricated from each anodized sample? I apologize if I have missed this but are the devices flexible? There is only one mention of this on line 104 "which were put into a glass capillary with electrolyte fully injected." What are the typical sizes of the glass capillary to accommodate the working and counter electrode?

Response to specific comment 1: Thanks the reviewer for the comment. All the data were collected using three devices for each anodized sample. This technique is repeatable and reliable. This device is composed of the flexible Ti wire and Pt wire substrate, so the whole device is flexible. The glass capillary has the diameter of 1.5 mm and length of 3 cm. All the information required by the reviewer was added in the experimental section colored in the yellow background as follows.

In lines 6-10 at page 6

“The glass capillary has the diameter of 1.5 mm and the length of 3 cm. This device is composed of the flexible Ti wire photoanode and the flexible Pt wire counter electrode. Both of the electrodes are flexible. However, the glass capillary is not flexible. By replacing the liquid electrode with the gel electrolyte and removing the glass capillary from the device, the whole device can thus be flexible.”

From the last line at page 6 to the line 1 at page 7

“All the data were collected using three devices for each anodized sample. This technique is repeatable and reliable.”

Comment 2: line 89: What is a "neutral cleaner"?

Response to specific comment 2: Thanks the reviewer for the comment. The neutral cleaner is used to wash the substrate. The main purpose is to remove the organic materials. The following sentence were also added in lines 6-7 at page 5 in the revised manuscript colored in the yellow background to explain the neutral cleaner.

“The neutral cleaner is used to wash the substrate via removing the organic materials.”

Comment 3: line 107: Is the hot adhesive "Surlyn" or have the authors used another sealant?

Response to specific comment 3: Thanks the reviewer for the comment. In this work we did not use Surlyn as the sealant. We put the photoanode and counter electrode in the glass capillary, and use a hot melt to seal both ends. The following sentence were also added in lines 13-14 at page 6 in the revised manuscript colored in the yellow background to explain the sealing process

“The hot melt adhesive was finally used to seal the tubular container at both ends for finishing the assembly of the FDSSC”.

Comment 4: lines 133-135: "The more charges generation may provide larger driving force for charge transfer and the recombination possibility may be reduced at the same time" Why would the recombination be reduced? Such specific reasoning as reduced recombination requires additional experiments as this is a huge issue with solar cells and simple "more charge generation" cannot be a viable explanation. Moreover, on the solar cell with 4 layers (Figure 1a, green line) the Jsc is significantly lower but the Voc remains the same. Therefore, this does not hold true in that case. I understand that the authors are speculating on the issue of increased Voc but they should be more careful with assumptions.

Response to specific comment 4: Thanks the reviewer for the suggestion. The reason for the increased VOC values may be due to the more charges accumulated in the conduction band of TiO2 with the thicker layers. The difference between the charge-accumulated conduction band edge of TiO2 and the redox potential of the electrolyte could be larger, and hence the VOC value could be large as well. The following sentence were also added in lines 1-4 at page 8 in the revised manuscript colored in the yellow background to explain the increased VOC values.

“The more charges could be accumulated in the conduction band of TiO2 with the thicker layers, so the difference between the charge-accumulated conduction band edge of TiO2 and the redox potential of the electrolyte could be larger”

Comment 5: line 154: What are "circular inks" Are they simply describing the morphology of the electrode after treatment?

Response to specific comment 5: Thanks the reviewer for the comment. Since the TNT has circular open ends, the removal of TNT would remain circular shape on the Ti wire. Hence, we used “circular inks” to describe the remaining shapes on the Ti wire. The following sentence were also added in lines 6-8 at page 10 in the revised manuscript colored in the yellow background to explain the term “circular inks”.

“Since the TNT has circular open ends, the removal of TNT would remain circular shape on the Ti wire. Hence, the term “circular inks” was used to describe the remaining shapes on the Ti wire.”

Comment 6: line 200-202: "higher JSC values were obtained for the FDSSC with the TNP/TNT TNT-printed Ti wire photoanodes than that for the cell with the TNP/Ti wire photoanode, owing to the TNT imprints on the substrate of the photoanode to provide better charge transfer." Why is that the explanation specifically? What proves the improved charge transfer? Is this some results from earlier work from the authors? If so a reference is needed. Higher Jsc could simply be from improved dye absorption (higher quantity of dye due to more surface area).

Response to specific comment 6: Thanks the reviewer for the suggestion. The better charge transfer can be inferred by the smaller charge-transfer resistance obtained from the Nyquist plot (Figure 5). Also, the higher dye adsorption is another reason for the high JSC value. Therefore, we modified the explanation in lines 5-8 at page 13 in the revised manuscript colored in the yellow background as follows.

“The higher JSC values were obtained for the FDSSC with the TNP/TNT-printed Ti wire photoanodes than that for the cell with the TNP/Ti wire photoanode, owing to the TNT imprints on the substrate of the photoanode to provide better charge transfer and larger surface area for dye adsorption.”

Comment 7: line 208-210: "Therefore, the best photovoltaic performance was obtained for the FDSSC with the photoanode prepared using the substrate pre-treated with the smallest anodization voltage."  The smallest anodization voltage measured here is zero (Fig 4) which does not show this behavior. Also since the authors see that the anodization voltage is inversely proportional to the efficiency, have they tried 20V, 10V? etc

Response to specific comment 7: Thanks the reviewer for the suggestion. The zero volt mentioned in Figure 4 indicates the sample without anodization, not the smallest anodization voltage. To avoid confusion, the following sentence were also added in lines 3-4 at page 13 in the revised manuscript colored in the yellow background to explain the “0 V”.

“The as-obtained Ti wire without anodization is indicated as Ti (0 V) in this figure.”

Also, the variation of the anodization voltage cannot only influence the diameter but also the depth of TNT inks. When using the voltage smaller than 30 V, nearly no inks can be observed in the SEM images. Therefore, 30 V is the smallest acceptable voltage to create TNT inks on the Ti wire. To explain this situation, the following sentences were added in lines 1-4 at the bottom at page 13 in the revised manuscript colored in the yellow background.

“The variation of the anodization voltage cannot only influence the diameter but also the depth of TNT inks. When using the anodization voltage smaller than 30 V, nearly no inks can be observed in the SEM images. Therefore, 30 V is the smallest acceptable anodization voltage to create TNT inks on the Ti wire in this study.”

Comment 8: Concerning the English language there are two main issues.

1) there are too many instances of the overusing the article "the" by the authors throughout the manuscript.

2) The authors tend to unnecessarily write extremely long sentences. There are several instances where the sentence is 6 or more lines long. This forces the interested reader to go back several times just to understand a simple point that the authors are trying to make. examples of unnecessarily long or confusing sentences include:

lines 27-33

lines 76-81

There are also bad sentences that definitely need rephrasing due to bad English or simplification such as

line 150-152

lines 249-252

Several simple corrections like this can greatly improve the interested reader's understanding of the authors' research.

Response to specific comment 8: Thanks the reviewer for the suggestion. 1) The term “the” was largely deleted in the revised manuscript. 2) The long sentences were modified to help reading. The modified parts were colored in the green backgrounds in the revised manuscript.

Comment 9: Other than those two main issues the manuscript is acceptable in English with minor spelling mistakes. (e.g. 1st line of the abstract: "deem-light". I am assuming that the authors mean dim-light? It would be easier for the scientific crowd to write "excellent low-light performance" which is something often used to characterize DSSCs) Unfortunately, there are several minor phrasing errors which can be easily avoided by a thorough proof reading by a native English speaker.

 Response to specific comment 9: Thanks the reviewer for the suggestion. The English writing of this manuscript was improved by a native English speaker to modify the errors.

Reviewer 3 Report

The present paper cannot be recommended for publication neither in Nanomaterials nor in any other scientific edition. Large additional job should be provided by the authors in experimental area and in arrangement of the material before their paper will be suitable for publishing. Even the paper is in general well written however it scientific soundness is rather low. Moreover a number of essential investigations which can increase possible contribution of current research is missing.

Authors claim the “A novel TiO2 nanotube (TNT) growth and removal”. However there is not prove of that NTs are really formed upon anodization at conditions used by authors. Line 154 reads that with anodization there are “circular inks on the surface”. But the NT formation was never proved. One or the explanation provided regarding the improved efficiency is the increase of the surface area, however the supporting experiments were not shown. Authors can use either BET or dye desorption methodology (first deposit dye on electrodes of same area and that wash it out with the same amount of solvent, and compare the absorption of the solvent solution). Thus the higher absorption of solvent used for higher surface area electrode can be justified. Authors several times mentioned the different numbers of NP coatings (Fig. 1) which give different NP layer thickness but those measurements were never presented. I mean the thickness of NP layer. Should be added to the text since this is the essential information for the paper. SEM figs are given with 100 um scale bar but authors claims that the “nanoparticles are redistributed” (Line 162). How is it possible to see NPs on images with such a low magnification? Images with higher magnification should be provided for more detail characterization of samples. Fig (b) shows that no TiO2 peaks are formed with different anodization voltages. This is weird since on Fig. (a) the TiO2 peaks upon anodization can be clearly seen. Moreover across the text authors several times claim that the anodization (at the same voltages) leads to the oxide formation, thus it is unclear why there are no trace of oxides in XRD (b) figure. Finally, best results were achieved for 30V anodization. Why did not authors try 25 or 20 V to find the optimum voltage? Also there are several English issues like line 150. “TNP deposited Ti wires”, Should be “coated”

Author Response

Reviewer #3

General comment: The present paper cannot be recommended for publication neither in Nanomaterials nor in any other scientific edition. Large additional job should be provided by the authors in experimental area and in arrangement of the material before their paper will be suitable for publishing. Even the paper is in general well-written however it scientific soundness is rather low. Moreover, a number of essential investigations which can increase possible contribution of current research is missing.

Response to the general comment: Thanks the reviewer for the comment. We have modified our manuscript thoroughly according to the reviewer’s comments carefully in the following specific comments.

Specific comments

Comment 1: Authors claim the “A novel TiO2 nanotube (TNT) growth and removal”. However, there is not prove of that NTs are really formed upon anodization at conditions used by authors. Line 154 reads that with anodization there are “circular inks on the surface”. But the NT formation was never proved.

Response to specific comment 1: Thanks the reviewer for the suggestion. The SEM image for the TNT was provided in the inserted figures in Figure 2(b)-(d) to confirm the formation of NT by using the anodization technique. The following sentence were added in lines 3-4 at page 10 in the revised manuscript colored in the yellow background to explain the formation of NT.

“The SEM images for the anodized Ti wire without removing the TNT were inserted in the corresponding figure in Figure 2(b)-(d).”

Comment 2: One or the explanation provided regarding the improved efficiency is the increase of the surface area, however the supporting experiments were not shown.

Response to specific comment 2: Thanks the reviewer for the suggestion. The increased surface area has been proved in our previous work, which was cited in ref. 24 in lines 8 and 14 at page 13 in the revised manuscript to support our explanation.

Comment 3: Authors can use either BET or dye desorption methodology (first deposit dye on electrodes of same area and that wash it out with the same amount of solvent, and compare the absorption of the solvent solution). Thus the higher absorption of solvent used for higher surface area electrode can be justified.

Response to specific comment 3: Thanks the reviewer for the comment. However, the surface area of the Ti wire used in this work is much smaller than the regular type of the photoanode, such as the FTO glass and Ti foil based photoanodes. The measurement of dye-adsorption would not be very accurate. Since the larger surface area has been proved in our previous work, we just cite our work in ref. 24 lines 8 and 14 at page 13 in the revised manuscript to support our explanation.

Comment 4: Authors several times mentioned the different numbers of NP coatings (Fig. 1) which give different NP layer thickness but those measurements were never presented. I mean the thickness of NP layer. Should be added to the text since this is the essential information for the paper.

Response to specific comment 4: Thanks the reviewer for the comment. The thickness of the TNP layer was estimated using the SEM images, as shown in Figure 1(a)-(e) in the revised manuscript. The following sentence were added in lines 9-12 at page 7 in the revised manuscript colored in the yellow background to explain the thickness of the TNP layers. The original Figure 1(a) and Figure 1(b) were moved to Figure 1(f) and Figure 1(g) in the revised manuscript.

“The thickness of the TNP layer was firstly examined using the SEM images. The side-view SEM images for the TNP/Ti photoanodes prepared using 1, 2, 3, 4 and 5 dip-coating layers were respectively shown in Figure 1(a)-(e). It was found that the thicker TNP layer can be obtained by using more dip-coating layers for fabricating the photoanodes.”

Comment 5: SEM figs are given with 100 um scale bar but authors claims that the “nanoparticles are redistributed” (Line 162). How is it possible to see NPs on images with such a low magnification? Images with higher magnification should be provided for more detail characterization of samples.

Response to specific comment 5: Thanks the reviewer for the comment. The high magnified SEM images for the TNP/Ti wire photoanodes were inserted in Figure 2(e)-(h) for closer observations. The following sentences were also added in lines 3-4 at page 10 in the revised manuscript colored in the yellow background to explain the inserted figures.

“The high magnifications for the SEM images were inserted in the corresponding figures.”

Comment 6: Fig (b) shows that no TiO2 peaks are formed with different anodization voltages. This is weird since on Fig. (a) the TiO2 peaks upon anodization can be clearly seen. Moreover, across the text authors several times claim that the anodization (at the same voltages) leads to the oxide formation, thus it is unclear why there are no trace of oxides in XRD (b) figure.

Response to specific comment 6: Thanks the reviewer for the comment. Sorry for the confusion. The Ti wire (Ti (0 V)) shown in Figure 3(b) is the pristine Ti wire, which is the same as “Ti wire” in Figure 3(a). Similarly, the TNT-printed Ti wires (Ti (30 V), Ti (40 V), and Ti (50 V)) shown in Figure 3(b) is the “sonicated anodized Ti wires” shown in Figure 3(a), not the “anodized Ti wire” in Figure 3(a) with TiO2 peaks.

Comment 7: Finally, best results were achieved for 30V anodization. Why did not authors try 25 or 20 V to find the optimum voltage?

Response to specific comment 7: Thanks the reviewer for the suggestion. The variation of the anodization voltage cannot only influence the diameter but also the depth of TNT inks. When using the voltage smaller than 30 V, nearly no inks can be observed in the SEM images. Therefore, 30 V is the smallest acceptable voltage to create TNT inks on the Ti wire. To explain this situation, the following sentences were added in lines 1-4 at the bottom at page 13 in the revised manuscript colored in the yellow background.

“The variation of the anodization voltage cannot only influence the diameter but also the depth of TNT inks. When using the anodization voltage smaller than 30 V, nearly no inks can be observed in the SEM images. Therefore, 30 V is the smallest acceptable anodization voltage to create TNT inks on the Ti wire in this study.”

Comment 8: Also there are several English issues like line 150. “TNP deposited Ti wires”, Should be “coated”

Response to specific comment 8: Thanks the reviewer for the suggestion. The English issue mentioned by this reviewer was modified and colored in the yellow background in the revised manuscript.

Round 2

Reviewer 3 Report

After revision, the manuscript is suitable for fabrication. However, I would like to recommend to authors to check English one more time since still there are some issues with English in the text.

See for example:““The thickness of the TNP layer was firstly examined using the SEM images.” “using” is not appropriate preposition here. “From” will sound better.

Or

Page 13. Lines 5-8. The sentence reads” The higher JSC values were obtained for the FDSSC with the TNP/TNT-printed Ti wire photoanodes than that for the cell with the TNP/Ti wire photoanode, owing to the TNT imprints on the substrate of the”.

“Than” sounds weird here. Something like “compared to” will be more suitable.

Author Response

Reviewer #3

General comment: After revision, the manuscript is suitable for fabrication. However, I would like to recommend to authors to check English one more time since still there are some issues with English in the text.

Response to the general comment: Thanks the reviewer for the comment. We have checked English throughout the manuscript.

Specific comments

Comment 1: See for example: “The thickness of the TNP layer was firstly examined using the SEM images.” “using” is not appropriate preposition here. “From” will sound better.

Response to specific comment 1: Thanks the reviewer for the suggestion. The term “using” was replaced by “from” in the sentence mentioned by the reviewer. Also, the similar words were replaced at page 11-15 colored in the yellow background in the revised manuscript.

Comment 2: Page 13. Lines 5-8. The sentence reads” The higher JSC values were obtained for the FDSSC with the TNP/TNT-printed Ti wire photoanodes than that for the cell with the TNP/Ti wire photoanode, owing to the TNT imprints on the substrate of the”. “Than” sounds weird here. Something like “compared to” will be more suitable.

Response to specific comment 2: Thanks the reviewer for the suggestion. The term “than” was replaced by “compared to” in the sentence mentioned by the reviewer. Also, the similar words were replaced at page 8, 13 and 20 colored in the yellow background in the revised manuscript.
